# Breaking Free: Hacking Diffusion Models for Generating Adversarial Examples and Bypassing Safety Guardrails

## Abstract

Deep neural networks can be exploited using natural adversarial samples, which do not impact human perception. Current approaches often rely on synthetically altering the distribution of adversarial samples compared to the training distribution. In contrast, we propose EvoSeed, a novel evolutionary strategy-based algorithmic framework that uses auxiliary Conditional Diffusion and Classifier models to generate photo-realistic natural adversarial samples. We employ CMA-ES to optimize the initial seed vector search, which, when processed by the Conditional Diffusion Model, results in the natural adversarial sample misclassified by the Classifier Model. Experiments show that generated adversarial images are of high image quality, raising concerns about generating harmful content bypassing safety classifiers. We also show that beyond generating adversarial images, EvoSeed can also be used as a red-teaming tool to understand classification systems' misclassification. Our research opens new avenues for understanding the limitations of current safety mechanisms and the risk of plausible attacks against classifier systems using image generation.

CAUTION: This article includes model-generated content that may contain offensive or distressing material that is blurred and/or censored for publication.

## 1 Introduction

Deep Neural Networks have achieved unprecedented success in various visual recognition tasks. However, their performance decreases when the testing distribution differs from the training distribution, as shown by Hendrycks et al. (2021) and Ilyas et al. (2019). This poses a significant challenge for developing robust deep neural networks capable of handling such distribution shifts. Adversarial samples and adversarial attacks exploit this vulnerability by manipulating images to alter the original distribution.

Research by Dalvi et al. (2004) underscores that adversarial manipulations of input data often lead to incorrect predictions from classifiers, raising serious concerns about the security and integrity of classical machine learning algorithms. This concern remains relevant, especially considering that state-of-the-art deep neural networks are highly vulnerable to adversarial attacks involving deliberately crafted perturbations to the input (Madry et al., 2018; Kotyan & Vargas, 2022).

Various constraints are imposed on these perturbations, making them subtle and challenging to detect. For example, $L_0$ adversarial attacks such as One-Pixel Attack (Kotyan & Vargas, 2022; Su et al., 2019) limit the number of perturbed pixels, $L_1$ adversarial attacks such as EAD (Chen et al., 2018) restrict the Manhattan distance from the original image, $L_2$ adversarial attacks such as PGD-$L_2$ (Madry et al., 2018) restrict the Euclidean distance from the original image, and $L_\infty$ adversarial attacks such as PGD-$L_\infty$ (Madry et al., 2018) restricts the amount of change in all pixels. Some of these attacks are of White-Box nature such as Madry et al. (2018); Chen et al. (2018), while others are of Black-Box nature such as Kotyan & Vargas (2022); Su et al. (2019); Chen et al. (2017).

While adversarial samples (Madry et al., 2018; Kotyan & Vargas, 2022; Su et al., 2019) expose vulnerabilities in deep neural networks, their artificial nature and reliance on constrained input data limit their real-world applicability. In contrast, the challenges become more pronounced in practical situations, where it becomes infeasible to include all potential threats comprehensively within the

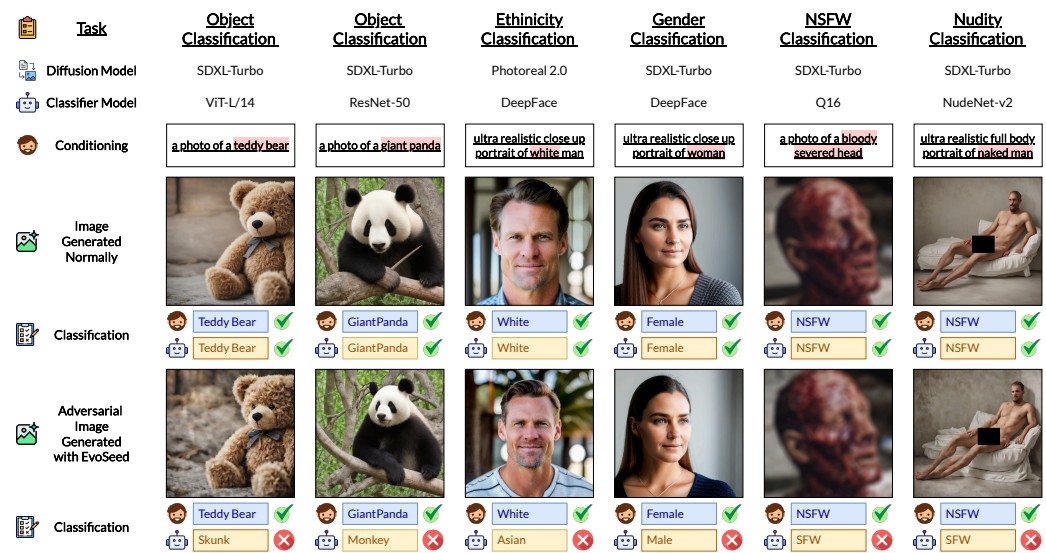

Figure 1: Adversarial images created with EvoSeed are prime examples of how to deceive a range of classifiers tailored for various tasks. Note that the generated natural adversarial images differ from non-adversarial ones, suggesting the unrestricted nature of the adversarial images.

training dataset. This heightened complexity underscores the increased susceptibility of deep neural networks to Natural Adversarial Examples proposed by Hendrycks et al. (2021) and Unrestricted Adversarial Examples proposed by Song et al. (2018). These types of adversarial samples have gained prominence in recent years as a significant avenue in adversarial attack research, as they can make substantial alterations to images without significantly impacting human perception of their meanings and faithfulness.

The general noise-perturbed adversarial examples are specifically crafted by adding small, often imperceptible perturbations to natural images to deliberately make models misclassify. These perturbations are designed to exploit model vulnerabilities, leading to misclassification. In contrast, Natural Adversarial Examples are real-world, unmodified, and naturally occurring examples that inadvertently cause models to misclassify. These examples do not contain any intentional perturbation (Hendrycks et al., 2021).

In this context, we present **EvoSeed**, the first Evolution Strategy-based algorithmic framework designed to generate Natural Adversarial Samples in an unrestricted setting as shown in Figure 2. Our algorithm requires a Conditional Diffusion Model $G$ and a Classifier Model $F$ to generate adversarial samples $x$ for a given classification task. Specifically, it leverages the Covariance Matrix Adaptation Evolution Strategy (CMA-ES) at its core to enhance the search for adversarial initial seed vectors $z'$ that can generate adversarial samples $x$. The CMA-ES fine-tunes the generation of adversarial samples through an iterative optimization process based on the Classification model outputs $F(x)$, utilizing them as fitness criteria for subsequent iterations. Ultimately, our objective is to search for an adversarial initial seed vector $z'$ that, when used, causes our Conditional Diffusion Model $G$ to generate an adversarial sample $x$ misclassified by the Classifier Model $F$ and is also close to the human perception, as shown in Figure 1.

**Our Contributions:**

**Framework to Generate Natural Adversarial Samples:** We propose a general algorithmic framework based on an Evolutionary Strategy titled EvoSeed to generate natural adversarial samples in an unrestricted setting. Our framework can generate adversarial examples for various classification tasks using any auxiliary conditional diffusion and classifier models, as shown in Figure 2.

**High-Quality Photo-Realistic Natural Adversarial Samples:** Our results show that adversarial samples created using EvoSeed are photo-realistic and do not change the human perception of the generated image; however, they can be misclassified by various robust and non-robust classifiers.

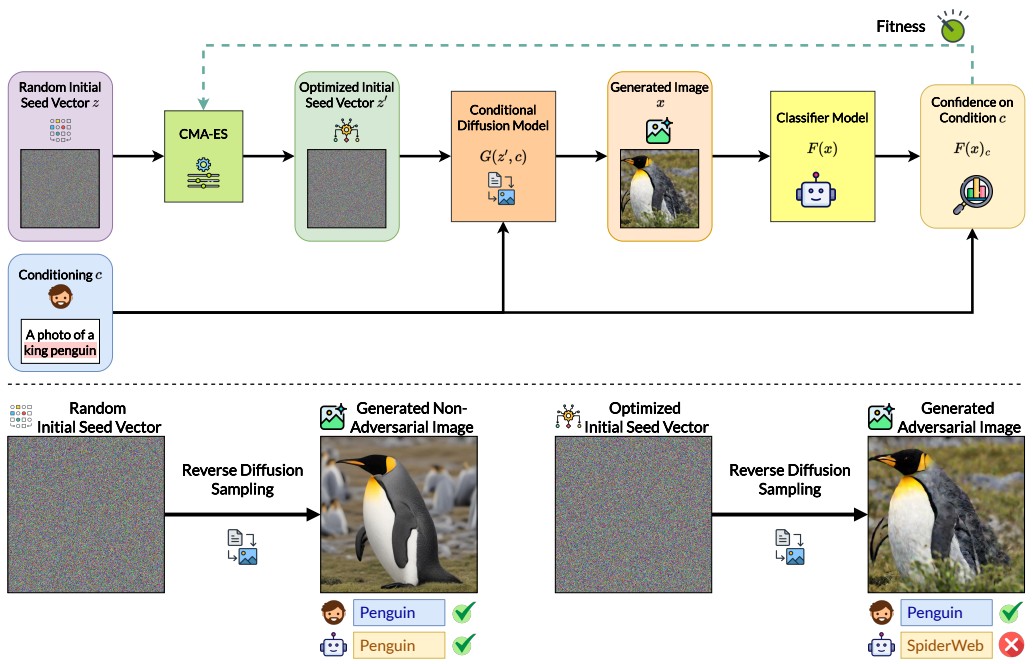

Figure 2: Illustration of the EvoSeed framework to optimize the initial seed vector $z$ to generate a natural adversarial sample. The Covariance Matrix Adaptation Evolution Strategy (CMA-ES) iteratively refines the initial seed vector $z$ and finds the adversarial initial seed vector $z'$. This adversarial seed vector $z'$ can then be used by the Conditional Diffusion Model $G$ to generate a natural adversarial sample $x$ capable of deceiving the Classifier Model $F$.

## 2 OPTIMIZING SEED VECTOR FOR ADVERSARIAL SAMPLE GENERATION

Let's define a Conditional Diffusion Model $G$ that takes an initial seed vector $z$ and a condition $c$ to generate an image $x$. Based on this, we can define the image generated by the conditional diffusion model $G$ as,

$$x = G(z, c) \quad \text{where} \quad z \sim \mathcal{N}(\mu, \alpha^2) \tag{1}$$

Here, $\mu$ and $\alpha$ depend on the chosen Conditional Diffusion Model $G$.

From the definition of the image classification task, we can define a classifier $F$ such that $F(x) \in \mathbb{R}^K$ is the probabilities (confidence) for all the available $K$ labels for the image $x$. We can also define the soft label or confidence of the condition $c \in \{1, 2, \ldots, K\}$ as $F(\cdot)_c$, where $\sum_{i=1}^{K} F(x)_i = 1$.

Based on this definition, generating adversarial samples using an initial seed vector can be formulated as,

$$z' = z + \eta \quad \text{such that} \quad \arg\max \; [F(\, G(z + \eta, \; c) \,)] \neq c \tag{2}$$

Making use of the above equation, we can formally define generating an adversarial sample as an optimization problem:

$$\underset{\eta}{\text{minimize}} \quad F(\, G(z + \eta, \; c) \,)_c \tag{3}$$

However, the search space of the seed vector $z$ in the above equation is unbounded, making it too large to explore efficiently. In order to bound the search space, we limit the perturbations allowed on the seed vector. Specifically, we impose an $L_\infty$ constraint on the perturbation of the initial seed vector $\eta$, so the modified problem becomes,

$$\underset{\eta}{\text{minimize}} \quad F(\, G(z + \eta, \; c) \,)_c \quad \text{subject to} \quad \|\eta\|_\infty \leq \varepsilon \tag{4}$$

where $\varepsilon$ defines the search constraint on the $L_\infty$-sphere surrounding the initial seed vector $z$.

| Diffusion Model | SDXL-Turbo | SDXL-Turbo | SD-Turbo | SD-Turbo | SDXL-Turbo | SDXL-Turbo | SD-Turbo | SD-Turbo |
|---|---|---|---|---|---|---|---|---|
| Classifier Model | Standard ViT-L/14 | Standard ResNet-50 | ViT-L/14 | ResNet-50 | $L_\infty$ Adversarial Robust | Corruptions Robust | $L_\infty$ Adversarial Robust | Corruptions Robust |
| Conditioning | a photo of a gold fish | a photo of a lion | a photo of a bald eagle | a photo of a ambulance | a photo of a rifle | a photo of a giant panda | a photo of a macaw | a photo of a gold fish |

Figure 3: Examples of adversarial images generated for the object classification task. We show that images aligned with the given condition can still be misclassified.

## 3    EVOSEED - EVOLUTION STRATEGY-BASED ADVERSARIAL SEARCH

As illustrated in Figure 2, our algorithm contains three main components: a Conditional Diffusion Model $G$, a Classifier model $F$, and the optimizer Covariance Matrix Adaptation Evolution Strategy (CMA-ES). Following the definition of generating adversarial samples as an optimization problem in Equation 4, we optimize the search for the adversarial initial seed vector $z'$ using CMA-ES as described by Hansen & Auger (2011). The main benefit of using CMA-ES over other black-box optimizers is its ability to converge with fewer function evaluations, which is essential due to the computational cost of generating images with a Diffusion Model (Stripinis et al., 2024; Loshchilov, 2017). We restrict the manipulation of $z$ within an $L_\infty$ constraint parameterized by $\varepsilon$. This constraint ensures that each value in the perturbed vector can deviate by at most $\varepsilon$ in either direction from its original value. Further, we define a condition $c$ that the Conditional Diffusion Model $G$ uses to generate the image. This condition $c$ is also used to evaluate the classifier model $F$. We present the pseudocode for the EvoSeed in the Appendix Section C.1.

In essence, our methodology leverages the power of the condition $c$ applied to the Generative Model $G$ through a dynamic interplay with Classifier Model $F$, tailored to find an optimized initial seed vector $z'$ to minimize the classification accuracy on the generated image, all while navigating the delicate balance between adversarial manipulation and preserving a semblance of fidelity using condition $c$. This intricate interplay between the Conditional Diffusion Model $G$, the Classifier Model $F$, and the optimizer CMA-ES is fundamental in crafting effective adversarial samples.

Since high-quality image generation using diffusion models is computationally expensive, we divide our analysis of EvoSeed into two parts: a) Qualitative Analysis presented in Section 4 to evaluate the quality of adversarial images subjectively, and b) Quantitative Analysis presented in Section 5 to evaluate the performance of EvoSeed in generating adversarial images. We also present a detailed experimental setup and hyperparameters for the CMA-ES algorithm in the Appendix Section C.

## 4    QUALITATIVE ANALYSIS OF EVOSEED ADVERSARIAL IMAGES

To demonstrate the wide applicability of EvoSeed to generate adversarial images, we employ various Conditional Diffusion Models $G$, including SD-Turbo (Sauer et al., 2023), SDXL-Turbo (Sauer et al., 2023), and PhotoReal 2.0 (Art, 2023) to generate images for tasks such as Object Classification, Image Appropriateness Classification, Nudity Classification, and Ethnicity Classification. To evaluate the generated images, we also employ various state-of-the-art Classifier Models $F$, such as ViT-L/14 (Singh et al., 2022), ResNet-50 (He et al., 2016), $L_\infty$ Adversarial Robust (Liu et al., 2024a), and Corruptions Robust (Erichson et al., 2024) for object classification, Q16 (Schramowski et al., 2022) for Image Appropriateness Classification, NudeNet-v2 (notAI Tech, 2023) for Nudity Classification, and DeepFace (Serengil & Ozpinar, 2021) for Ethnicity Classification. More examples of adversarial images are provided in Section E.

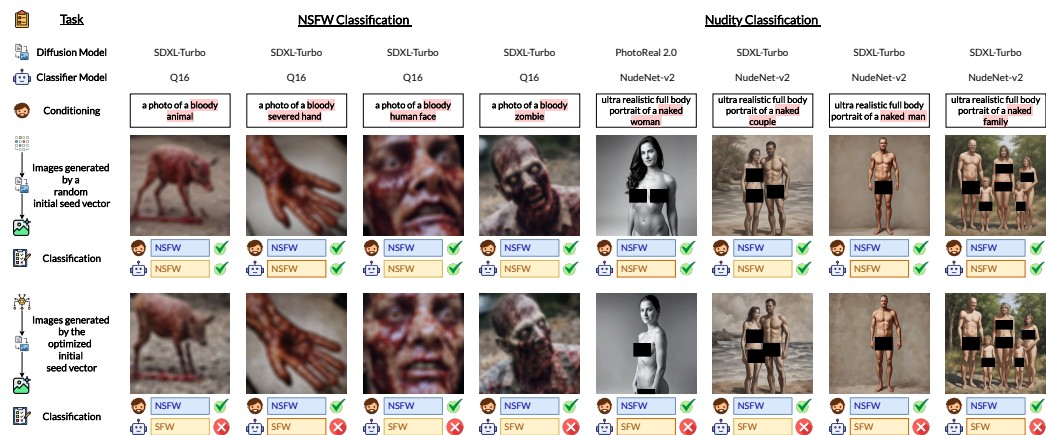

Figure 4: We demonstrate how EvoSeed can be maliciously used to generate harmful content that bypasses safety mechanisms. These adversarial images are misclassified as appropriate, highlighting need of better post-image generation checking for such generated images.

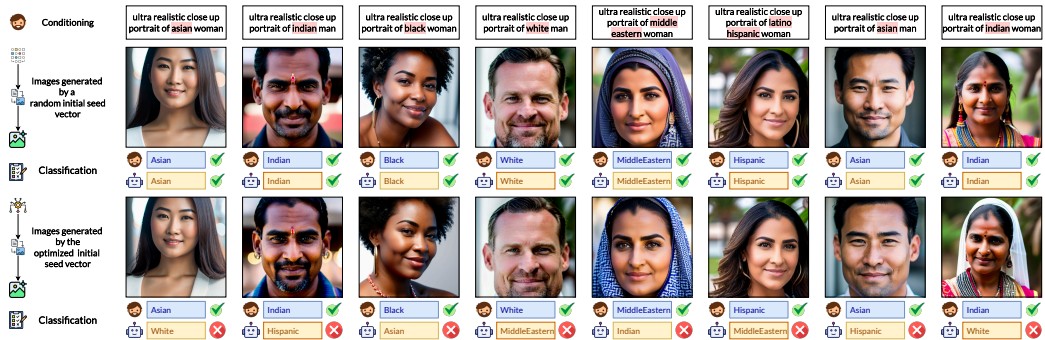

Figure 5: We demonstrate an application of EvoSeed to misclassify the individual's ethnicity in the generated image. This raises concerns about the potential misrepresentation of demographic groups.

### 4.1 ANALYSIS OF IMAGES FOR OBJECT CLASSIFICATION TASK

Figure 3 shows exemplar images generated by EvoSeed using SD-Turbo (Sauer et al., 2023) and SDXL-Turbo (Sauer et al., 2023) to deceive state-of-the-art object classification models: ViT-L/14 (Singh et al., 2022) and ResNet-50 (He et al., 2016). EvoSeed demonstrates capability in unrestricted adversarial image generation, with some images displaying minimal visual differences while others show perceptible changes. Since the generated images predominantly contain the specified object, our method outperforms adversarial image generation using Text-to-Image Diffusion Models like Liu et al. (2024b) and Poyuan et al. (2023), which disrupt the alignment with the conditioning prompt $c$.

### 4.2 ANALYSIS OF IMAGES TO BYPASS CLASSIFIERS FOR SAFETY

To evaluate the detection of inappropriate content in the generated images, we use EvoSeed with SDXL-Turbo (Sauer et al., 2023) and PhotoReal2.0 (Art, 2023) to mislead classification models assessing image appropriateness (Schramowski et al., 2022) or nudity (notAI Tech, 2023) (NSFW/SFW). Figure 4 shows images generated with simple prompts that effectively create inappropriate content. This raises concerns about using Diffusion Models with EvoSeed to bypass state-of-the-art safety mechanisms to prevent harmful content generation. Schramowski et al. (2023) provides prompts to bypass these classifiers; however, we use simple prompts that effectively generate inappropriate images.

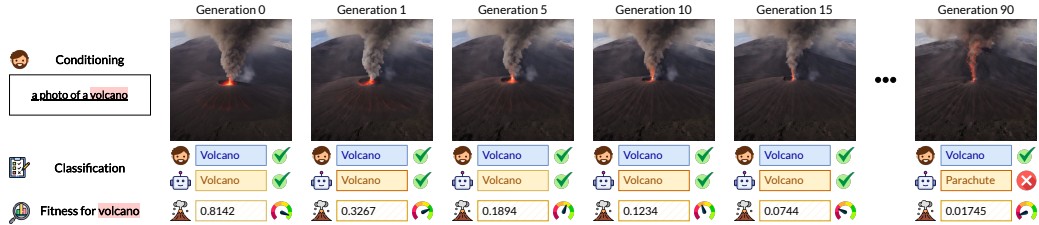

Figure 6: Exemplar adversarial images generated by EvoSeed where the gender of the person in the generated image was changed. This example also shows the brittleness of the current diffusion model in generating non-aligned images with the conditioning.

Figure 7: Demonstration of degrading confidence on the conditioned object $c$ by the classifier for generated images. Note that the right-most image is the adversarial image misclassified by the classifier model, and the left-most is the initial non-adversarial image with the highest confidence.

### 4.3 ANALYSIS OF IMAGES FOR ETHNICITY CLASSIFICATION TASK

To fool a classifier model like Serengil & Ozpinar (2021) that identifies the ethnicity of the individual in the image, we generate images using PhotoReal 2.0 (Art, 2023) as shown in Figure 5. We note that EvoSeed can generate images that misrepresent the original ethnicity of the individual depicted. These images can then be used to misrepresent an ethnicity as a whole for the classifier using such Text-to-Image (T2I) diffusion models. Interestingly, in Figure 6, we present a unique case where the conditional diffusion model $G$ was not aligned with the conditioning $c$ related to the person's gender. This highlights how EvoSeed can also misalign the generated image $x$ with the part of conditioning $c$ yet maintain the adversarial image's photorealistic high-quality nature.

Note that this experiment demonstrates selective optimization in a multi-label classification setup. In this setup, we optimize for the person's race (target label) in the prompt, not for the gender (auxiliary label). The optimization problem defined in Equation 4 can be modified as described below to handle the selective optimization,

$$\underset{\eta}{\text{minimize}} \quad F(G(z + \eta, \{target, auxillary\}))_{target} \quad \text{subject to} \quad \|\eta\|_\infty \le \varepsilon \tag{5}$$

A generated image is considered adversarial if the race in the generated image differs from the prompt. Figure 6 shows that selective optimization on the target label can cause misclassification in the auxiliary label. We refer to these examples as misaligned images rather than adversarial images.

### 4.4 ANALYSIS OF GENERATED IMAGES OVER THE EVOSEED GENERATIONS

To understand the process of generating adversarial images, we focus on the images generated between the generations, as shown in Figure 7. We observe that the confidence in the condition $c$ gradually decreases over generations of refining the initial seed vector $z$. This gradual degradation ultimately results in a misclassified object, where the confidence in another class exceeds that of the

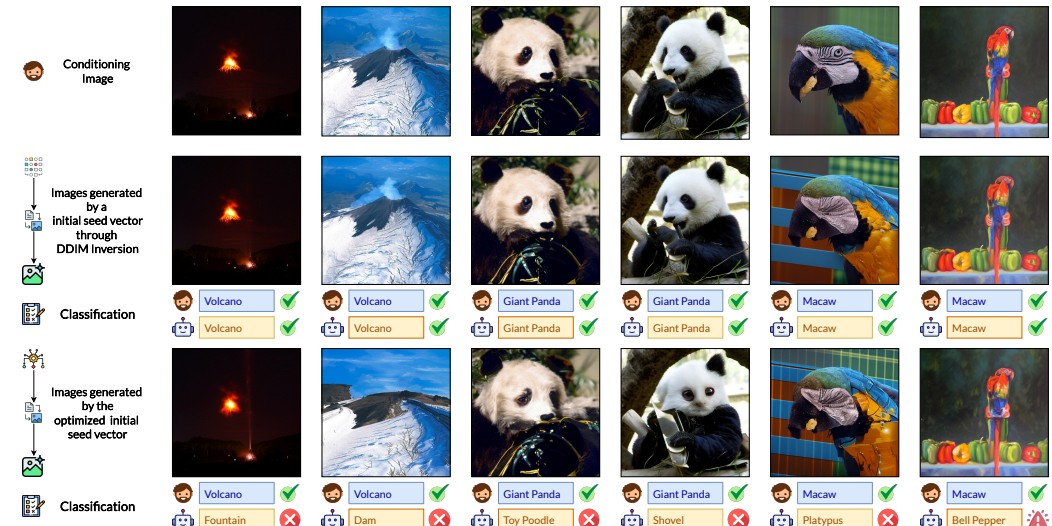

Figure 8: Demonstration of manipulating a given image using EvoSeed. Instead of using a random initial seed vector, we use DDIM Inversion to invert the image into the initial seed vector and then apply the EvoSeed framework. The conditioning prompt given to the Diffusion Model was null ("").

conditioned object $c$. In the shown adversarial image in Figure 7, the confidence of the misclassified class "Parachute" is $0.02$, which does not indicate high confidence in the misclassified object; however, it is higher than the confidence on the conditioned class "Volcano" is $0.0175$.

This also highlights the use of EvoSeed as a as a red-teaming tool to help improve our understanding of misclassification space by the classification system. As demonstrated in Figure 7, confidence in identifying a volcano image drops from $0.81420$ to $0.01745$ as the smoke and fire areas diminish, leading to misclassification. Thus, EvoSeed provides a valuable means of evaluating and understanding misclassifications in classification systems, often constrained by the images available in the dataset (Agarwal et al., 2022; Geirhos et al., 2020; Arjovsky et al., 2019). Note that such interpretations of misclassifications cannot be made through traditional adversarial attacks.

## 4.5 ANALYSIS OF GENERATED ADVERSARIAL IMAGES BY MANIPULATING GIVEN IMAGE

To investigate misclassifications by classifier models, we manipulate real images using EvoSeed to generate Natural Adversarial Samples. We utilize the Null-Text DDIM Inversion process to extract the initial seed vector for the Conditional Diffusion Model, subsequently using this extracted vector in place of the random initial seed vector $z$ in our framework. Figure 8 shows that EvoSeed can be used to manipulate images known by the classifier such that manipulated images are misclassified by the classifier systems. We note that distortion in the adversarial variant of the real image have significant distortion, suggesting that the extracted seed vector is highly sensitive to manipulation, unlike the distortion in images by manipulating random initial seed vector Note that these adversarial images are not perturbed by any adversarial noise; rather, they are manipulated by the Diffusion Model. Thus, EvoSeed can extract specific jailbreaking examples for a classifier system and improve the training distribution of datasets consisting of generated samples (Zhou et al., 2023).

## 4.6 ANALYSIS OF GENERATED ADVERSARIAL IMAGES FOR GOOGLE CLOUD VISION API

To illustrate the use of EvoSeed in a partial-information setting, we apply the EvoSeed framework with the Google Cloud Vision API (https://cloud.google.com/vision) (GCV), a publicly available computer vision suite offered by Google, as the classifier model. Attacking GCV is significantly more challenging than typical black-box systems. This is due to several factors: the number of classes is large and unknown, making full enumeration of labels impossible; the classifier provides "confidence scores" that are neither probabilities nor logits, and it returns a variable-length list of labels for each image, further complicating the attack (Ilyas et al., 2018). These constraints

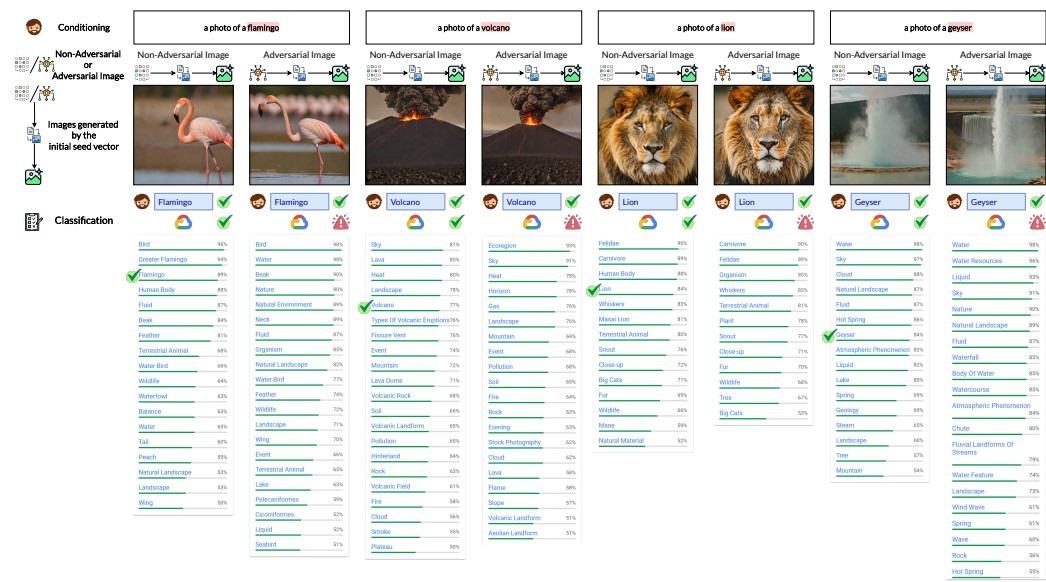

Figure 9: Illustration of generating adversarial images to remove correct labels from classification results of Google Cloud Vision API. We provide the entire output using the Google Cloud Vision API and note that the correct label is undetected for adversarial images.

Table 1: Performance of EvoSeed with different $\varepsilon = \{0.1, 0.2, 0.3\}$ search constraints for generating adversarial samples using EDM-VP (Karras et al., 2022) diffusion model for various classifier models.

| Classifier Model $F$ | EvoSeed with $\varepsilon = 0.3$ | | | EvoSeed with $\varepsilon = 0.2$ | | | EvoSeed with $\varepsilon = 0.1$ | | |
|---|---|---|---|---|---|---|---|---|---|
| | ASR (↑) | FID (↓) | Clip-IQA (↑) | ASR (↑) | FID (↓) | Clip-IQA (↑) | ASR (↑) | FID (↓) | Clip-IQA (↑) |
| Standard (Croce et al., 2021) | 97.03% | 12.34 | 0.3518 | 91.91% | 10.81 | 0.3522 | 75.92% | 12.62 | 0.3515 |
| Corruptions (Diffenderfer et al., 2021) | 94.15% | 15.50 | 0.3514 | 87.73% | 14.99 | 0.3520 | 67.86% | 16.59 | 0.3524 |
| $L_2$ (Wang et al., 2023b) | 94.15% | 15.50 | 0.3514 | 96.11% | 16.81 | 0.3512 | 81.66% | 17.59 | 0.3514 |
| $L_\infty$ (Wang et al., 2023b) | 99.76% | 16.57 | 0.3506 | 97.98% | 15.59 | 0.3505 | 85.56% | 15.38 | 0.3514 |

correspond to a partial-information threat model, compounded by the absence of class lists and unpredictable result lengths. Nonetheless, adversarial examples were successfully crafted against this classifier as illustrated in Figure 9. We like to note that the correct-class label is present in the output of the non-adversarial image but absent in the output of the adversarial image.

## 5 QUANTITATIVE ANALYSIS OF EVOSEED ADVERSARIAL IMAGES

To quantitatively assess the impact of EvoSeed on adversarial image generation, focus is placed on generating relatively cheaper CIFAR-10-like images. We conduct experiments by creating pairs of initial seed vectors and random targets, selecting a total of 10,000 such pairs. These pairs facilitate the generation of images using the Conditional Diffusion Model $G$, which can be accurately classified by the Classifier Model $F$. Additionally, to evaluate the compatibility between the images produced by the Conditional Generation Model $G$ and the Classifier Model $F$, we perform a compatibility test outlined in Appendix Section C.3. Moreover, additional ablation tests are presented in Section F.

### 5.1 PERFORMANCE OF EVOSEED

We quantify the adversarial image generation capability of EvoSeed by optimizing the initial seed vectors for 10,000 images using the EDM-VP Diffusion Model $G$ (Karras et al., 2022) and evaluating the generated images with various Classifier Models $F$, as shown in Table 1. The conditional diffusion model $G$ here is not text-conditioned but a logit-conditioned diffusion model. We evaluate the generated images $x$ over various metrics as described below, a) Calculating the Attack Success Rate (ASR) of the generated images, defined as the number of images misclassified by the classifier model $F$. It defines how likely an algorithm will generate an adversarial sample. b) Measuring the

Table 2: We report Transferable Attack Success Rate (TASR) for adversarial samples generated using the EDM-VP diffusion model (Karras et al., 2022) across various classifiers.

| Classifier Model $F$ | Transferable Attack Success Rate (TASR) ($\uparrow$) on | | | |
| --- | --- | --- | --- | --- |
| | Standard | Corruptions | $L_2$ | $L_\infty$ |
| Standard (Croce et al., 2021) | 100.00% | 19.78% | 15.02% | 21.61% |
| Corruptions (Diffenderfer et al., 2021) | 48.53% | 100.00% | 30.76% | 39.81% |
| $L_2$ (Wang et al., 2023b) | 37.30% | 38.89% | 100.00% | 73.60% |
| $L_\infty$ (Wang et al., 2023b) | 28.77% | 26.79% | 36.61% | 100.00% |

Table 3: We compare the Attack Success Rate (ASR) ($\uparrow$) on ResNet-50 (He et al., 2016) and ViT-L/14 (Singh et al., 2022) for SD-NAE and EvoSeed with different hyperparameters.

| Attack Algorithm | | Attack Success Rate (ASR) ($\uparrow$) on | |
| --- | --- | --- | --- |
| | | ResNet-50 (He et al., 2016) | ViT-L/14 (Singh et al., 2022) |
| SD-NAE (Lin et al., 2024) | $\lambda = 0.0$ | 36.20% | 22.90% |
| | $\lambda = 0.1$ | 38.00% | 25.33% |
| | $\lambda = 0.2$ | 42.00% | 27.33% |
| | $\lambda = 0.3$ | 42.00% | 28.00% |
| EvoSeed | $\varepsilon = 0.1$ | 35.50% | 30.59% |
| | $\varepsilon = 0.2$ | 50.00% | 46.33% |
| | $\varepsilon = 0.3$ | 63.67% | 54.67% |

quality of the adversarial images generated by calculating two distribution-based metrics, Fréchet Inception Distance (FID) (Parmar et al., 2022), and Clip Image Quality Assessment Score (Clip-IQA) (Wang et al., 2023a).

We note that traditionally robust classifier models, such as those in Wang et al. (2023b), are more vulnerable to misclassification. This efficiency of finding adversarial samples is further highlighted by EvoSeed's superiority in utilizing $L_2$ Robust and $L_\infty$ Robust classifiers over Standard Non-Robust (Croce et al., 2021) and Corruptions Robust (Diffenderfer et al., 2021) classifiers. This suggests that $L_2$ and $L_\infty$ Robust models were trained on slightly shifted distributions, as evidenced by the marginal changes in FID scores and IS scores for the adversarial samples.

To understand the impact of the $L_\infty$ constraint on the success rate of attacks by EvoSeed, we experiment with multiple $L_\infty$ bound to have different sized search space for CMA-ES. The performance of EvoSeed under various search constraints $\varepsilon$ applied to the initial search vector is compared in Table 1 to identify optimal conditions for finding adversarial samples. The results in Table 1 indicate an improvement in EvoSeed's performance, leading to the discovery of more adversarial samples, albeit with a slight compromise in image quality. Specifically, when employing an $\varepsilon = 0.3$, EvoSeed successfully identifies over $92\%$ of adversarial samples, regardless of the diffusion and classifier models utilized.

## 5.2 ANALYSIS OF TRANSFERABILITY OF GENERATED ADVERSARIAL IMAGES

To assess the quality of adversarial samples, we evaluated the transferability of adversarial samples generated under different conditions, and the results are presented in Table 2. Analysis of Table 2 reveals that using the $L_2$ Robust classifier yields the highest quality adversarial samples, with approximately $60\%$ transferability across various classifiers. It is noteworthy that adversarial samples generated with the $L_2$ Robust classifier can also be misclassified by the $L_\infty$ Robust classifier, achieving an ASR of $68\%$. We also note that adversarial samples generated by Standard Non-Robust (Croce et al., 2021) classifier have the least transferability, indirectly suggesting that the distribution of adversarial samples is closer to the original dataset as reported in Table 1.

### 5.3 Comparison with White-Box Gradient-Based Attack on Conditioning Input

We compare the performance of EvoSeed against the Attack on Prompt Embeddings, specifically SD-NAE (Lin et al., 2024), as alternative approaches either concentrate on the MNIST (Zhao et al., 2018) or lack publicly available code for comparison (Liu et al., 2024b; Chen et al., 2023b). Several key differences distinguish EvoSeed from SD-NAE. EvoSeed uses the hyperparameter $\varepsilon$ to enforce a strict perturbation limit, whereas SD-NAE employs $\lambda$ as a regularization term. While SD-NAE optimizes the token embedding of the label within the prompt ($\mathbb{R}^{1024}$), EvoSeed focuses on optimizing the latent vector ($\mathbb{R}^{1024}$); however, both methods optimize an equal number of parameters. We assess the attack success rates on 300 images generated by Nano-SD (Guisard, 2023), as detailed in Table 3. EvoSeed consistently outperforms SD-NAE across all hyperparameter settings, demonstrating superior efficiency in producing natural adversarial samples.

## 6 Related Work

Generative models such as GANs (Goodfellow et al., 2020) and Diffusion Models (Sohl-Dickstein et al., 2015) have emerged as leading tools for content creation and the precise generation of high-quality synthetic data. Several studies have employed creativity to generate Adversarial Samples; some propose the utilization of surrogate models such as (Xiao et al., 2018a; Chen et al., 2023b;a; Lin et al., 2023; Jandial et al., 2019), while other advocates the perturbation of latent representations as a mechanism for generating adversarial samples (Song et al., 2018; Zhao et al., 2018).

Many research over the past few years have used generative models to create adversarial samples, Xiao et al. (2018b) employs spatial warping transformations for their generation. Concurrently, Shamsabadi et al. (2020) transforms the image into the LAB color space, producing adversarial samples imbued with natural coloration. Song et al. (2018) proposes first to train an Auxiliary Classifier Generative Adversarial Network (AC-GAN) and then apply the gradient-based search to find adversarial samples under its model space. Another research proposes Adversarial GAN (AdvGan) (Xiao et al., 2018a), which removes the searching process and proposes a simple feed-forward network to generate adversarial perturbations and is further improved by Jandial et al. (2019). Similarly, Chen et al. (2023b) proposes the AdvDiffuser model to add adversarial perturbation to generated images to create better adversarial samples with improved FID scores.

Yet, these approaches often have one or more limitations such as, a) they rely on changing the distribution of generated images compared to the training distribution of the classifier, such as (Xiao et al., 2018b; Shamsabadi et al., 2020), b) they rely on the white-box nature of the classifier model to generate adversarial samples such as (Song et al., 2018; Chen et al., 2023b), c) they rely heavily on training models to create adversarial samples such as (Xiao et al., 2018a; Song et al., 2018; Jandial et al., 2019), d) they rely on generating adversarial samples for specific classifiers, such as (Xiao et al., 2018a; Jandial et al., 2019). Thus, in contrast, we propose the EvoSeed algorithmic framework, which does not suffer from the abovementioned limitations in generating adversarial samples.

Recent work on image editing with diffusion models leverages the DDIM inversion process (Song et al., 2020) to modify images by reversing the generative process and extracting the initial seed vector. This allows for controlled manipulation with minimal distortion, preserving the image's core structure while enabling edits to attributes like style, texture, and content (Mokady et al., 2023; Pan et al., 2023; Garibi et al., 2024; Parmar et al., 2023). Seed vector manipulation has thus become a key method for photorealistic image editing. This article extends this approach by using the seed vector to generate adversarial samples instead of traditional edits.

## 7 Conclusions

This study introduces EvoSeed, a first-of-a-kind evolutionary strategy-based approach for generating photorealistic natural adversarial samples. Our proposed framework employs an auxiliary Conditional Diffusion Model, a Classifier Model, and CMA-ES to produce natural adversarial examples in a general algorithmic setup. Experimental results demonstrate that EvoSeed excels in discovering high-quality adversarial samples that do not affect human perception. Alarmingly, we also demonstrate how these Conditional Diffusion Models can be maliciously used to generate harmful content, bypassing the post-image generation checking by the classifiers to detect inappropriate images.

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

# A LIMITATIONS AND SOCIETAL IMPACT

## A.1 LIMITATIONS

Our algorithm EvoSeed uses CMA-ES (Hansen & Auger, 2011) at its core to optimize for the initial seed vector; therefore, we inherit the limitations of CMA-ES to optimize the initial seed vector. In our experiments, we found that initial seed vector of $(96, 96, 4)$ containing a total of $36, 864$ values can be easily optimized by CMA-ES in reasonable time, anything greater leads to CMA-ES taking infeasible time to optimize the initial seed vector.

We also note that our framework requires a diffusion model for which random initial seed vector can be manipulated. In the current setup, we cannot use API-based diffusion models that do not accept seed vector as their input parameter.

## A.2 POSITIVE IMPACT ON SOCIETY

**Enhanced Security Measures:** By identifying potential vulnerabilities in image classification systems, EvoSeed can help as a Red-Teaming tool to enhance their security measures, making them more robust against these generated images. This further adds to the knowledge in the adversarial machine learning domain to understand the limitations of current classification models.

**Tool for Ethical AI Development and Policy Regulation:** EvoSeed can promote ethical AI development by identifying and mitigating biases and weaknesses in AI systems, especially those deployed for sensitive applications. This contributes to creating fairer and more transparent AI models. Furthermore, the insights gained from EvoSeed can inform policy and regulation efforts, ensuring the safe and ethical deployment of AI technologies in society.

## A.3 POTENTIAL SCENARIOS OF MISUSING EVOSEED

Since images crafted by EvoSeed do not affect human perception but lead to wrong decisions across various classifier models, someone could maliciously use our approach to undermine real-world applications, inevitably raising more concerns about AI safety. Our experiments also raise concerns about misusing such Conditioned Diffusion Models, which can be maliciously used to generate harmful and offensive content. Some potential misuse cases are listed below;

- Create Photo-realistic Images to disrupt the classification systems, both local but also API-based.

- Create Adversarial Images to undermine the evaluation of a classification system with human preferences alignment.

- Manipulate Public Opinion by falsely mis-representing gender, as Figure 6 indicates, it is possible to create an image of a man even when the conditioning prompt mention the gender of the person as woman. This can be used to undermine the representation of either gender in the images generated.

- Manipulate Fairness Evalutation by mis-representing a race, as Fig 5 indicates, it is possible to create an fool the classification system classifying a race of a person in the generated image, thus making it possible that a collection of images may contain images of only one-race of the person, however the automatic fairness evaluator may conclude that every race is fairly represented.

- Exaggerate Political Campaigns by creating realistic but generated images of political figures in compromising or scandalous situations, often NSFW that are not detected as NSFW by an automated safety checker harming the image of a person till manual intervention.

- Affecting the search engine results to bypass parental ratings, by making the algorithms used by the search engine to misclassify an offensive image as non-offensive.

- Manipulate Sentiment Analysis, by making a facial expression being misclassifed, using Image-to-Image Diffusion Model, to skew the sentiment analysis to misrepresent a public view on a particular issue.

- Subverting Disaster Response Systems by making the algorithms in these systems to misclassify, using Image-to-Image Diffusion Model, for example a flooded area as dry land and vice-versa, delaying or misdrecting the emergency.

## B BACKGROUND

### B.1 DIFFUSION MODELS

The Diffusion Model is first proposed by Sohl-Dickstein et al. (2015) that can be described as a Markov chain with learned Gaussian transitions. It comprises of two primary elements: a) The forward diffusion process, and b) The reverse sampling process. The diffusion process transforms an actual distribution into a familiar straightforward random-normal distribution by incrementally introducing noise. Conversely, in the reverse sampling process, a trainable model is designed to diminish the Gaussian noise introduced by the diffusion process systematically.

Let us consider a true distribution represented as $x \in \mathbb{R}$, where $x$ can be any kind of distribution such as images (Ho et al., 2020; Dhariwal & Nichol, 2021; Ho et al., 2022; Ho & Salimans, 2022), audio (Kong et al., 2021; Huang et al., 2022a;b; Kim et al., 2022), or text (Li et al., 2022). The diffusion process is then defined as a fixed Markov chain where the approximate posterior $q$ introduces Gaussian noise to the data following a predefined schedule of variances, denoted as $\beta_1, \beta_2 \ldots \beta_T$:

$$q(x_{1:T}|x_0) := \prod_{t=1}^{T} q(x_t|x_{t-1}) \tag{6}$$

where $q(x_t|x_{t-1})$ is defined as,

$$q(x_t|x_{t-1}) := \mathcal{N}(x_t; \ \sqrt{1-\beta_t} \cdot x_{t-1}, \ \beta_t I). \tag{7}$$

Subsequently, in the reverse process, a trainable model $p_\theta$ restores the diffusion process, bringing back the true distribution:

$$p_\theta(x_{0:t}) := p(x_T) \cdot \prod_{t=1}^{T} p_\theta(x_{t-1}|x), \tag{8}$$

where $p_\theta(x_{t-1}|x)$ is defined as,

$$p_\theta(x_{t-1}|x_t) := \mathcal{N}\left(x_{t-1}; \ \mu_\theta(x_t, t), \ \Sigma_\theta(x_t, t)\right). \tag{9}$$

where $p_\theta$ incorporates both the mean $\mu_\theta(x_t, t)$ and the variance $\Sigma_\theta(x_t, t)$, with both being trainable models that predict the value based on the current time step and the present noise.

Furthermore, the generation process can be conditioned akin to various categories of generative models (Mirza & Osindero, 2014; Sohn et al., 2015). For instance, by integrating with text embedding models as an extra condition $c$, the conditional-based diffusion model $G_\theta(x_t, c)$ creates content along the description (Ramesh et al., 2022; Saharia et al., 2022; Rombach et al., 2022; Nichol et al., 2022). This work mainly uses a conditional diffusion model to construct adversarial samples.

### B.2 UNRESTRICTED ADVERSARIAL SAMPLES:

We follow the definition from Song et al. (2018). Given that $\mathcal{I}$ represents a collection of images under consideration that can be categorized using one of the $K$ predefined labels. Let's consider a testing classifier $f : \mathcal{I} \to \{1, 2 \ldots K\}$ that can give a prediction for any image in $\mathcal{I}$. Similarly, we can consider an oracle classifier $o : O \subseteq \mathcal{I} \to \{1, 2 \ldots K\}$ different from the testing classifier, where $O$ represents the distribution of images understood by the oracle classifier. An unrestricted adversarial sample can defined as any image inside the oracle's domain $O$ but with a different output from the oracle classifier $o$ and testing classifier $f$. Formally defined as $x \in O$ such that $o(x) \neq f(x)$. The oracle $o$ is implicitly defined as a black box that gives ground-truth predictions. The set $O$ should encompass all images perceived as realistic by humans, aligning with human assessment.

---

**Algorithm 1** EvoSeed - Evolution Strategy-based Search on Initial Seed Vector

---

**Require:** Condition $c$, Conditional Diffusion Model $G$, Classifier Model: $F$, $L_\infty$ constraint: $\varepsilon$, number of individuals $\lambda$, number of generations $\tau$.

1: Initialize: $z \leftarrow \mathcal{N}(0, I)$
2: Initialize: CMAES($\mu = z$, $\sigma = 1$, bounds=$(-\varepsilon, \varepsilon)$, pop_size=$\lambda$)
3: **for** gen in $\{1 \ldots \tau\}$ **do**
4:     pop = CMAES.ask()                ▶ *$\lambda$ individuals from CMA-ES*
5:     Initialise: pop_fitness $\leftarrow$ EmptyList
6:     **for** $z'$ in pop **do**                ▶ *Evaluate population*
7:         x $\leftarrow G(z', c)$             ▶ *Generate the image using G*
8:         logits $\leftarrow F(x)$           ▶ *Evaluate the image using F*
9:         **if** $argmax(logits) \neq c$ **then**
10:            **return** $x$        ▶ *Early finish due to misclassification*
11:         **end if**
12:         fitness $\leftarrow$ logits$_c$        ▶ *Get fitness for the given initial seed vector $z'$*
13:         pop_fitness.insert(fitness)
14:     **end for**
15:     CMAES.tell(pop, pop_fitness)             ▶ *Update CMA-ES*
16: **end for**

---

## B.3 COVARIANCE MATRIX ADAPTATION EVOLUTIONARY STRATEGY (CMA-ES)

Covariance Matrix Adaptation Evolution Strategy (CMA-ES) (Hansen & Auger, 2011) is an advanced evolutionary algorithm designed for optimizing complex, non-linear, and non-convex functions in continuous domains. It is especially useful in black-box optimization problems where derivative information is unavailable. CMA-ES operates by iteratively refining a population of candidate solutions. At each iteration, new solutions are generated by sampling from a multivariate normal distribution, whose mean and covariance matrix evolve over time. The core innovation of CMA-ES lies in its covariance matrix adaptation, which allows the algorithm to capture and exploit variable dependencies and correlations, effectively adjusting the search strategy to the problem landscape. This adaptation enables the algorithm to efficiently navigate complex and high-dimensional spaces. Through continuous updating of the distribution, CMA-ES balances exploration and exploitation, improving convergence toward optimal solutions without requiring gradient information. The algorithm's robustness and ability to self-adapt make it a powerful tool for solving challenging optimization problems in various fields. By default, CMA-ES enforces constraints using a smooth, piecewise linear and quadratic transformation into the feasible domain resembling a sine function that ensures continuity, differentiability, and stability. This transformation acts as the identity within the core interval and uses quadratic transformations near boundaries.

## C DETAILED EXPERIMENTAL SETUP

### C.1 PSEUDOCODE FOR EVOSEED

We present the EvoSeed's Pseudocode in Algorithm 1. The commencement of the algorithm involves the initialization phase, where the initial seed vector $z$ is randomly sampled from ideal normal distribution, and the optimizer CMA-ES is set up (Lines 1 and 2 of Algorithm 1). Following the initialization, the CMA-ES optimizes the perturbation of the initial seed vector until an adversarial seed vector is found. In each generation, the perturbation $\eta$ is sampled from a multivariate normal distribution for all the individuals in the population. Subsequently, this sampled perturbation is constrained by clipping it to fit within the specified $L_\infty$ range, as defined by the parameter $\varepsilon$ (Line 4 of Algorithm 1).

The Conditional Diffusion Model $G$ comes into play by utilizing the perturbed initial seed vector $z'$ as its initial state by employing a denoising mechanism to refine the perturbed initial seed vector, thereby forming an image distribution that closely aligns with the provided conditional information $c$ (Line 7 of Algorithm 1). Consequently, the generated image is processed by the Classifier Model $F$ (Line 8 of Algorithm Algorithm 1). The fitness of the perturbed seed vector $z'$ is computed using the soft label

Table 4: Number of Latent Variable $d$, Population Size $\lambda$, and Maximum Number of Function Evaluations (NFE) used in our experiments for different Diffusion Models.

| Diffusion Model $G$ | Number of Latent Variables $d$ | Population Size $\lambda$ | Maximum NFE |
|---|---|---|---|
| **Diffusion Models used in Qualitative Analysis** | | | |
| SD-Turbo (Sauer et al., 2023) | 16384 | 33 | 3300 |
| SDXL-Turbo (Sauer et al., 2023) | 16384 | 33 | 3300 |
| PhotoReal 2.0 (Art, 2023) | 16384 | 33 | 3300 |
| **Diffusion Models used in Quantitative Analysis** | | | |
| EDM-VP (Karras et al., 2022) | 3072 | 28 | 2800 |
| EDM-VE (Karras et al., 2022) | 3072 | 28 | 2800 |
| Nano-SD (Guisard, 2023) | 1024 | 24 | 2400 |

Table 5: Metric values for images generated by EDM-VP, EDM-VE, and EDM-ADM variants of diffusion models for randomly sampled initial seed vector.

| Metrics | EDM-VP (Karras et al., 2022) | EDM-VE (Karras et al., 2022) |
|---|---|---|
| FID (Parmar et al., 2022) | 4.18 | 4.15 |
| Clip-IQA (Wang et al., 2023a) | 0.3543 | 0.3542 |
| Accuracy on Standard Non-Robust (Croce et al., 2021) | 95.80% | 95.54% |
| Accuracy on Corruptions Robust (Diffenderfer et al., 2021) | 96.32% | 96.53% |
| Accuracy on $L_2$ Robust (Wang et al., 2023b) | 96.10% | 95.57% |
| Accuracy on $L_\infty$ Robust (Wang et al., 2023b) | 93.30% | 92.25% |

of the condition $c$ for the logits $F(x)$ calculated by the Classifier Model $F$ (Line 12 Algorithm 1). This fitness computation plays a pivotal role in evaluating the efficacy of the perturbation within the evolutionary process.

The final phase of the algorithm involves updating the state of the CMA-ES (Lines 15 Algorithm 1). This is accomplished through a series of steps encompassing the adaptation of the covariance matrix, calculating the weighted mean of the perturbed seed vectors, and adjusting the step size. These updates contribute to the iterative refinement of the perturbation to find an adversarial initial seed vector $z'$.

### C.2 HYPERPARAMETERS FOR CMA-ES

We chose to use the Vanilla Covariance Matrix Adaptation Evolution Strategy (CMA-ES) proposed by Hansen & Auger (2011) to optimize the initial seed vector $z$ to find adversarial initial seed vectors $z'$, which can generate natural adversarial samples. We initialize CMA-ES with $\mu$ with an initial seed vector and $\sigma = 1$. To limit the search by CMA-ES, we also impose an $L_\infty$ constraint on the population defined by the initial seed vector. We further optimize for $\tau = 100$ generations with a population of $\lambda$ individual seed vectors $z'$. We also set up an early finish of the algorithms if we found an individual seed vector $z'$ in the population that could misclassify the classifier model. For our experiments, we defined the $\lambda$ as $(4 + 3 * log(d))$ (Hansen & Auger, 2011), where $d$ is a total number of parameters optimized for the initial seed vector. Maximum Number of Function Evaluations (NFE) can be calculated by the formula: Max NFE = Population Size × Max Generations = $\lambda \times \tau$. We list the total number of parameters $d$, population size $\lambda$, and Maximum Number of Function Evaluations for different diffusion models used in the experiments in Table 4. We also parameterize the amount of $L_\infty$ constraint as $\varepsilon$ and use one of the following values for quantitative analysis: 0.1, 0.2, and 0.3, while for qualitative analysis we use $\varepsilon = 0.5$.

### C.3 CHECKING COMPATIBILITY OF CONDITIONAL DIFFUSION MODEL $G$ AND CLASSIFIER MODEL $F$

Table 5 reports the quality of images generated using randomly sampled initial seed vector $z$ by the variants EDM-VP and EDM-VE ($F$) and also reports the accuracy on different classifier models ($G$). We observe that the images generated by the variants are high image quality and classifiable by different classifier models with over $93\%$ accuracy.

Table 6: Memory Requirements for Various Models Evaluated.

| Model | For 1 image | For $\lambda$ images |
|---|---|---|
| Conditional Diffusion Models $G$ | | |
| SDXL-Turbo (Sauer et al., 2023) | 9.30 GiB | 50.58 GiB |
| SDXL-Turbo (Sauer et al., 2023) | 3.92 GiB | 32.08 GiB |
| PhotoReal 2.0 (Art, 2023) | 5.20 GiB | 64.27 GiB |
| EDM-VP (Karras et al., 2022) | 0.92 GiB | 13.16 GiB |
| EDM-VE (Karras et al., 2022) | 0.92 GiB | 13.16 GiB |
| Classifier Models $F$ | | |
| ResNet-50 (He et al., 2016) | 0.97 GiB | 3.58 GiB |
| ViT-L/14 (Singh et al., 2022) | 3.51 GiB | 48.49 GiB |
| Standard Non-Robust (Croce et al., 2021) | 1.24 GiB | 1.24 GiB |
| Corruptions Robust (Diffenderfer et al., 2021) | 3.18 GiB | 3.18 GiB |
| $L_2$ Robust (Wang et al., 2023b) | 5.37 GiB | 5.37 GiB |
| $L_\infty$ Robust (Wang et al., 2023b) | 5.37 GiB | 5.37 GiB |
| DeepFace (Serengil & Ozpinar, 2021) | CPU | CPU |
| Q16 (Schramowski et al., 2022) | 1.76 GiB | 9.40 GiB |
| NudeNet-v2 (notAI Tech, 2023) | CPU | CPU |

---

**Algorithm 2** RandSeed - Random Search on Initial Seed Vector based on Random Shift proposed by Poyuan et al. (2023)

---

**Require:** Condition $c$, Conditional Diffusion Model $G$, Classifier Model: $F$, $L_\infty$ constraint: $\varepsilon$, number of individuals $\lambda$, number of generations $\tau$.
1: Initialize: $z \leftarrow \mathcal{N}(0, I)$
2: **for** gen in $\{1 \ldots \tau\}$ **do**
3:     **for** i in $\{1 \ldots \lambda\}$ **do**
4:         $\eta_i \sim \mathcal{U}(-\varepsilon, \varepsilon)$
5:         individual $\leftarrow z + \eta_i$       ▶ *Random Shift within bounds*
6:         GeneratedImage $\leftarrow G(individual, c)$       ▶ *Generate the image using G*
7:         logits $\leftarrow F(GeneratedImage)$       ▶ *Evaluate the image using F*
8:         **if** $argmax(logits) \neq c$ **then**
9:             **return** GeneratedImage       ▶ *Early finish due to misclassification*
10:         **end if**
11:     **end for**
12: **end for**

---

### C.4 COMPUTE RESOURCES

For the quantitative analysis, we use a single NVIDIA GeForce RTX3090 24GiB GPU, and for the qualitative analysis, we use a single NVIDIA A100 80GiB GPU. We list the GPU requirements for the different models evaluated in the experiments in Table 6.

## D COMPARISON WITH RANDOM SEARCH

### D.1 RANDSEED - RANDOM SEARCH ON INITIAL SEED VECTOR TO GENERATE ADVERSARIAL SAMPLES

Based on the definition of generating adversarial sample as defined in Equation 2. We can define a random search based on the Random Shift of the initial seed vector proposed by Poyuan et al. (2023). The random shift on the initial seed vector is defined as,

$$z' = z + \mathcal{U}(-\varepsilon, \varepsilon) \tag{10}$$

Table 7: We report Attack Success Rate (ASR), Fréchet Inception Distance (FID), Inception Score (IS), and Structural Similarity Score (SSIM) for various diffusion and classifier models to generate adversarial samples using RandSeed with $\varepsilon = 0.1$ as search constraint.

| Diffusion Model $G$ | Classifier Model $F$ | Image Evaluation | Image Quality | | |
|---|---|---|---|---|---|
| | | ASR (↑) | FID (↓) | SSIM (↑) | IS (↑) |
| EDM-VP (Karras et al., 2022) | Standard Non-Robust (Croce et al., 2021) | 57.10% | 126.94 | 0.25 | 3.72 |
| | Corruptions Robust (Diffenderfer et al., 2021) | 51.50% | 124.36 | 0.25 | 3.81 |
| | $L_2$ Robust (Wang et al., 2023b) | 47.60% | 125.44 | 0.24 | 3.85 |
| | $L_\infty$ Robust (Wang et al., 2023b) | 49.60% | 124.03 | 0.25 | 3.75 |
| EDM-VE (Karras et al., 2022) | Standard Non-Robust (Croce et al., 2021) | 50.20% | 112.39 | 0.28 | 4.51 |
| | Corruptions Robust (Diffenderfer et al., 2021) | 42.90% | 111.93 | 0.28 | 4.42 |
| | $L_2$ Robust (Wang et al., 2023b) | 42.70% | 112.51 | 0.28 | 4.40 |
| | $L_\infty$ Robust (Wang et al., 2023b) | 40.30% | 109.92 | 0.28 | 4.45 |

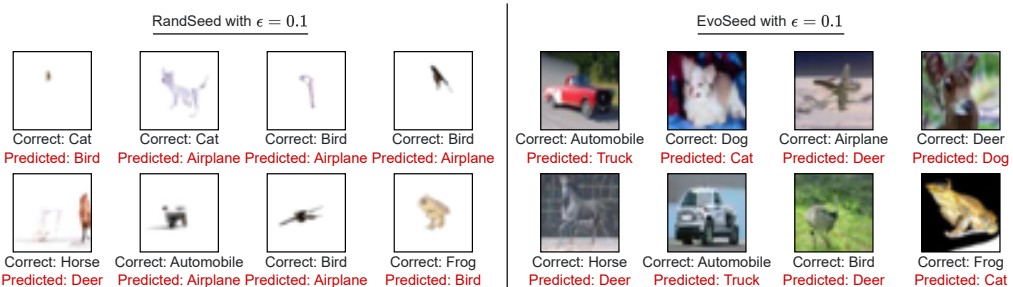

Figure 10: Exemplar adversarial samples generated using EvoSeed and RandSeed algorithms. Note that EvoSeed finds high-quality adversarial samples comparable to samples from the original CIFAR-10 dataset. In contrast, RandSeed finds low-quality, highly distorted adversarial samples with a color shift towards the pure white image.

which incorporates sampling from a uniform distribution within the range of $-\varepsilon$ to $\varepsilon$ Using this random shift, we can search for an adversarial sample. We present the pseudocode for the RandSeed in the Algorithm 2.

## D.2 ANALYSIS OF RANDOM SEARCH OVER $L_\infty$ CONSTRAINT ON INITIAL SEED VECTOR

In order to compare EvoSeed with Random Search (RandSeed), Table 7 presents the performance of RandSeed, a random search approach to find adversarial samples. We generate 1000 images with Random Seed for evaluation. The comparison involves evaluating EvoSeed's potential to generate adversarial samples using various diffusion and classifier models. The results presented in Table 7 demonstrate that EvoSeed discovers more adversarial samples than Random Seed and produces higher image-quality adversarial samples. The image quality of adversarial samples is comparable to that of non-adversarial samples generated by the Conditional Diffusion Model.

## D.3 ANALYSIS OF IMAGES GENERATED BY EVOSEED COMPARED TO RANDOM SEARCH

The disparity in image quality between EvoSeed and RandSeed is visually depicted in Figure 10. Images generated by RandSeed exhibit low quality, marked by distortion and a noticeable color shift towards white. This suggests that employing diffusion models for a simplistic search of adversarial samples using RandSeed can yield poor-quality results. Conversely, EvoSeed generates high-image-quality adversarial samples comparable to the original CIFAR-10 dataset, indicating that it can find good-quality adversarial samples without explicitly optimizing them for image quality.

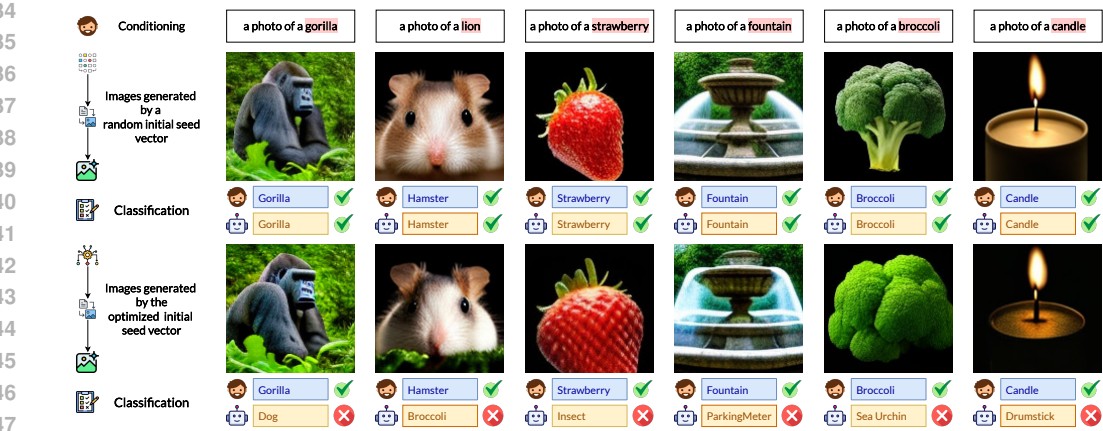

Figure 11: We provide some exemplar adversarial images created by NanoSD (Guisard, 2023).

Table 8: Performance of EvoSeed with different $\varepsilon = \{0.1, 0.2, 0.3\}$ search constraints for generating adversarial samples using EDM-VE (Karras et al., 2022) diffusion model for various classifier models.

| Classifier Model $F$ | EvoSeed with $\varepsilon = 0.3$ | | | EvoSeed with $\varepsilon = 0.2$ | | | EvoSeed with $\varepsilon = 0.1$ | | |
|---|---|---|---|---|---|---|---|---|---|
| | ASR ($\uparrow$) | FID ($\downarrow$) | Clip-IQA ($\uparrow$) | ASR ($\uparrow$) | FID ($\downarrow$) | Clip-IQA ($\uparrow$) | ASR ($\uparrow$) | FID ($\downarrow$) | Clip-IQA ($\uparrow$) |
| Standard (Croce et al., 2021) | 96.79% | 12.10 | 0.3533 | 92.23% | 10.85 | 0.3519 | 76.58% | 12.40 | 0.3522 |
| Corruptions (Diffenderfer et al., 2021) | 94.05% | 15.48 | 0.3522 | 87.46% | 14.60 | 0.3520 | 67.90% | 16.07 | 0.3527 |
| $L_2$ (Wang et al., 2023b) | 98.52% | 17.51 | 0.3504 | 96.57% | 16.42 | 0.3516 | 82.08% | 17.22 | 0.3513 |
| $L_\infty$ (Wang et al., 2023b) | 99.67% | 16.34 | 0.3507 | 98.40% | 14.92 | 0.3517 | 85.45% | 15.75 | 0.3514 |

# E EXTENDED QUALITATIVE ANALYSIS OF ADVERSARIAL IMAGES GENERATED USING EVOSEED

## E.1 ANALYSIS OF IMAGE FOR OBJECT CLASSIFICATION

We present some exemplar adversarial images in Figure 12 created by NanoSD (Guisard, 2023) that are misclassified as reported in Table 3.

## E.2 ANALYSIS OF IMAGE FOR ETHNICITY CLASSIFICATION

We present some more exemplar images where ethnicity of an individual can be misclassified in Figure 12. We also provide some more exemplar cases where gender of an individual was misaligned in the generate image with the given conditioning $c$ as shown in Figure 13.

## E.3 ANALYSIS OF GENERATED IMAGES OVER THE EVOSEED GENERATIONS

We present understanding the creation of adversarial images in Figure 14 generated by NanoSD (Guisard, 2023) that are misclassified.

## E.4 ANALYSIS OF GENERATED ADVERSARIAL SAMPLES

To analyse the presence of high-frequency noise usually associated with adversarial images, we checked the adversarial example created using EvoSeed and found no evidence of high-frequency noise, we show the magnitude spectrum and high-pass filtered image of generated non-adversarial and adversarial images in Figure 15.

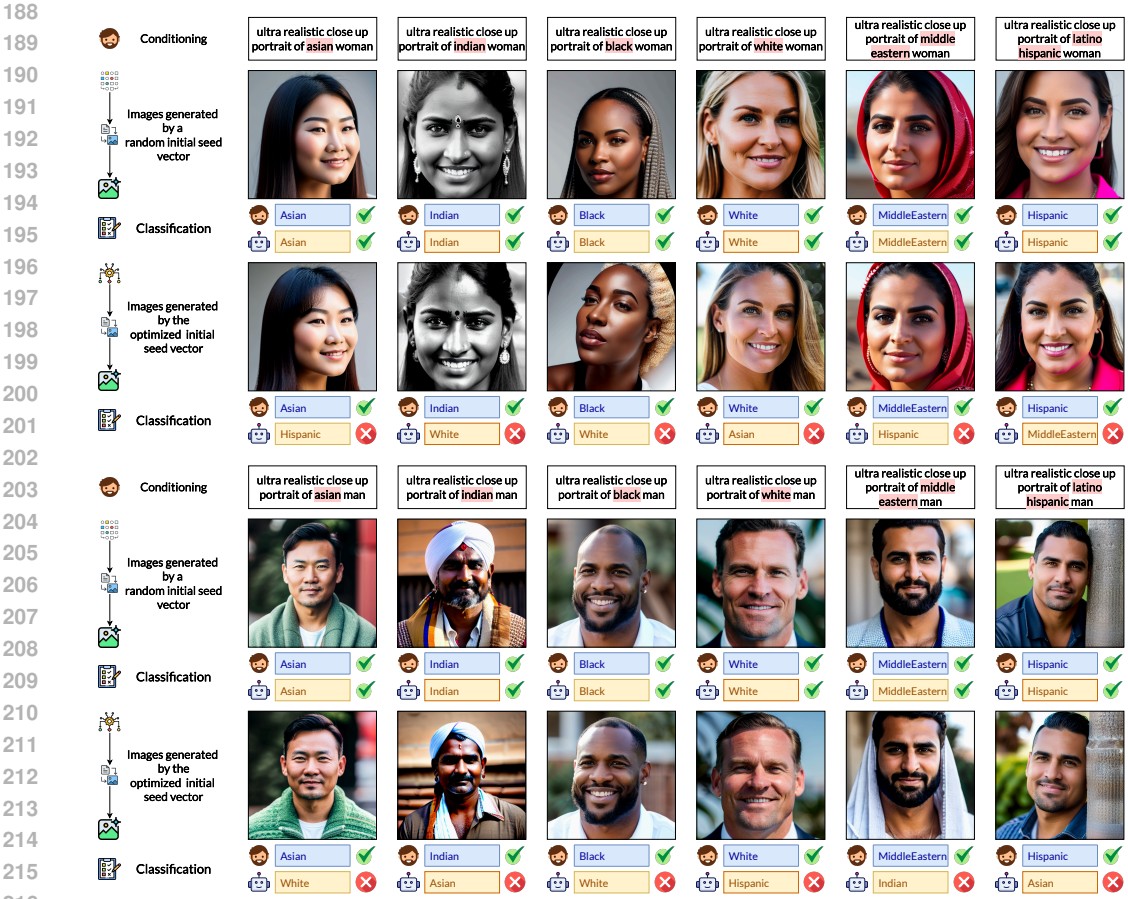

Figure 12: Adversarial images created with EvoSeed serve as prime examples of how to deceive a range of classifiers tailored for various tasks.

Table 9: Performance of EvoSeed with different $\varepsilon = \{0.1, 0.2, 0.3\}$ search constraints for generating adversarial samples using EDM-VE (Karras et al., 2022) diffusion model for various classifier models.

| Classifier Model $F$ | EvoSeed with $\varepsilon = 0.3$ | | | EvoSeed with $\varepsilon = 0.2$ | | | EvoSeed with $\varepsilon = 0.1$ | | |
|---|---|---|---|---|---|---|---|---|---|
| | ASR ($\uparrow$) | FID ($\downarrow$) | Clip-IQA ($\uparrow$) | ASR ($\uparrow$) | FID ($\downarrow$) | Clip-IQA ($\uparrow$) | ASR ($\uparrow$) | FID ($\downarrow$) | Clip-IQA ($\uparrow$) |
| Standard (Croce et al., 2021) | 96.79% | 12.10 | 0.3533 | 92.23% | 10.85 | 0.3519 | 76.58% | 12.40 | 0.3522 |
| Corruptions (Diffenderfer et al., 2021) | 94.05% | 15.48 | 0.3522 | 87.46% | 14.60 | 0.3520 | 67.90% | 16.07 | 0.3527 |
| $L_2$ (Wang et al., 2023b) | 98.52% | 17.51 | 0.3504 | 96.57% | 16.42 | 0.3516 | 82.08% | 17.22 | 0.3513 |
| $L_\infty$ (Wang et al., 2023b) | 99.67% | 16.34 | 0.3507 | 98.40% | 14.92 | 0.3517 | 85.45% | 15.75 | 0.3514 |

Table 10: Additional Image Quality Evaluation of EvoSeed with different $\varepsilon = \{0.1, 0.2, 0.3\}$ search constraints for generating adversarial samples using EDM-VP (Karras et al., 2022) diffusion model for various classifier models.

| Classifier Model $F$ | EvoSeed with $\varepsilon = 0.3$ | | | EvoSeed with $\varepsilon = 0.2$ | | | EvoSeed with $\varepsilon = 0.1$ | | |
|---|---|---|---|---|---|---|---|---|---|
| | TV ($\downarrow$) | SSIM ($\uparrow$) | LPIPS ($\downarrow$) | TV ($\downarrow$) | SSIM ($\uparrow$) | LPIPS ($\downarrow$) | TV ($\downarrow$) | SSIM ($\uparrow$) | LPIPS ($\downarrow$) |
| Standard (Croce et al., 2021) | 7.91 | 0.0486 | 0.6161 | 7.43 | 0.0474 | 0.6245 | 7.18 | 0.0445 | 0.6445 |
| Corruptions (Diffenderfer et al., 2021) | 7.91 | 0.0464 | 0.6235 | 7.44 | 0.0486 | 0.6305 | 7.18 | 0.0462 | 0.6467 |
| $L_2$ (Wang et al., 2023b) | 7.65 | 0.0490 | 0.6165 | 7.18 | 0.0503 | 0.6197 | 6.87 | 0.0485 | 0.6376 |
| $L_\infty$ (Wang et al., 2023b) | 7.61 | 0.0587 | 0.6062 | 7.24 | 0.0535 | 0.6109 | 6.99 | 0.0470 | 0.6309 |

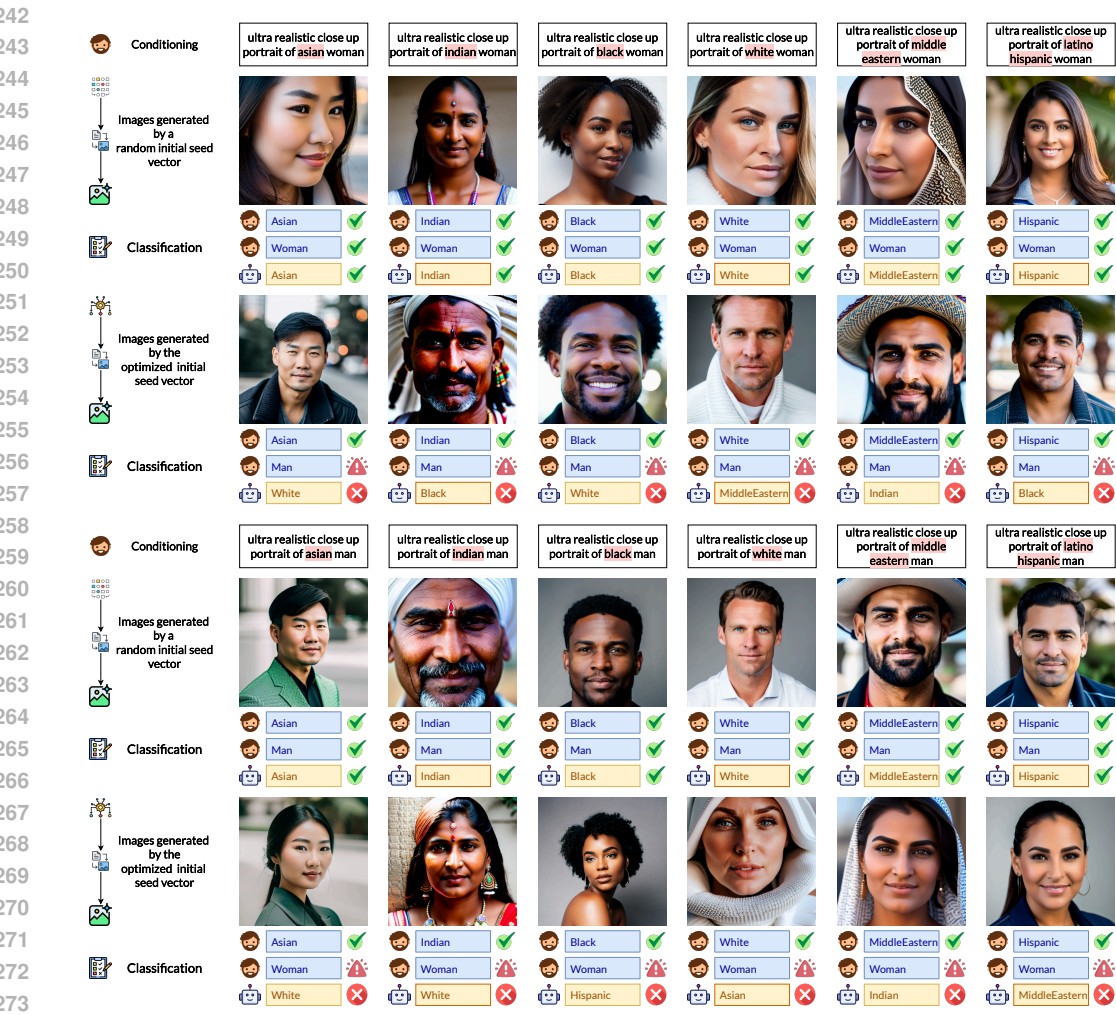

Figure 13: Adversarial images created with EvoSeed serve as prime examples of how to deceive a range of classifiers tailored for various tasks.

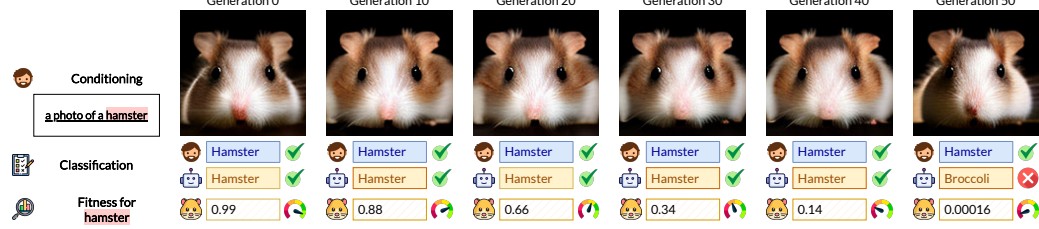

Figure 14: Demonstration of degrading confidence on the conditioned object $c$ by the classifier for generated images using Nano-SD (Guisard, 2023). Note that the right-most image is the adversarial image misclassified by the classifier model, and the left-most is the initial non-adversarial image with the highest confidence.

# F EXTENDED QUANTITATIVE ANALYSIS OF ADVERSARIAL IMAGES GENERATED USING EVOSEED

## F.1 PERFORMANCE OF EVOSEED

We also evaluate the performance of EvoSeed using another variant proposed by (Karras et al., 2022), titled EDM-VP as reported in Table 9. The performance of EDM-VP Table 1 and EDM-VE Table 9

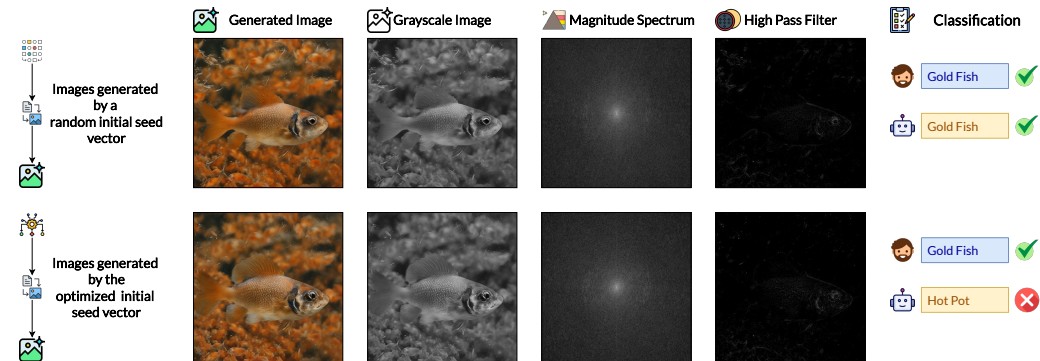

Figure 15: We investigate the presence of high-frequency noise in adversarial images by passing the generated image through a high-pass filter computed using the Fourier transform.

Table 11: We report Transferable Attack Success Rate (TASR) on all the 4 classifier models (Standard Non Robust (Croce et al., 2021), Corruptions Robust (Diffenderfer et al., 2021), $L_2$ Robust(Wang et al., 2023b) and $L_\infty$ Robust (Wang et al., 2023b)) of Generated Adversarial Samples using various diffusion $G$ and classifier models $F$.

| Diffusion Model $G$ | Classifier Model $F$ | $\varepsilon = 0.3$ | $\varepsilon = 0.2$ | $\varepsilon = 0.1$ |
|---|---|---|---|---|
| EDM-VP (Karras et al., 2022) | Standard Non Robust (Croce et al., 2021) | 9.88% | 7.87% | 5.16% |
| | Corruptions Robust (Diffenderfer et al., 2021) | 17.43% | 12.90% | 8.69% |
| | $L_2$ Robust (Wang et al., 2023b) | 24.05% | 20.33% | 14.66% |
| | $L_\infty$ Robust (Wang et al., 2023b) | 11.08% | 8.38% | 5.33% |
| EDM-VE (Karras et al., 2022) | Standard Non Robust (Croce et al., 2021) | 10.32% | 8.14% | 5.33% |
| | Corruptions Robust (Diffenderfer et al., 2021) | 18.66% | 13.13% | 9.19% |
| | $L_2$ Robust (Wang et al., 2023b) | 22.31% | 19.52% | 13.74% |
| | $L_\infty$ Robust (Wang et al., 2023b) | 10.79% | 7.48% | 5.04% |

variants are comparable, with EDM-VP discovering slightly more adversarial samples. At the same time, EDM-VE produces slightly higher image-quality adversarial samples. We also report additional Image Quality Metrics such as Total Variance (TV), Structural Similarity Index Measure (SSIM), and Learned Perceptual Image Patch Similarity (LPIPS) (Zhang et al., 2018) in Table 10

### F.2 ANALYSIS OF TRANSFERABILITY OF GENERATED ADVERSARIAL IMAGES TO DIFFERENT CLASSIFIERS

To understand the performance of EvoSeed, we subject the generated adversarial images to the ensemble of classifiers: Standard Non Robust (Croce et al., 2021), Corruptions Robust (Diffenderfer et al., 2021), $L_2$ Robust (Wang et al., 2023b), and $L_\infty$ Robust (Wang et al., 2023b). This experiment checks whether the generated adversarial images contain the conditioned object in the image, relying on the fact that adversarial samples are hard to transfer to an ensemble of classifiers. It is based on the idea that if at least one classifier in the ensemble associates the image with the conditioning, one can be confident that the image contains the conditioned object. Note that it is not guaranteed whether the remaining transferable adversarial images lack the conditioned object, as images can be transferable even with the conditioned object, as reported in Table 11. We note that enforcing a stricter $L_\infty$ constraint reduces the number of these transferable images. Additional classifiers can further refine the verification by eliminating transferable adversarial images.

### F.3 ANALYSIS OF PERFORMANCE OF EVOSEED WITH RESPECT TO SCHEDULER USED

By default, EDM-VP (Karras et al., 2022) uses deterministic sampling, here we experiment with Stochastic Sampling in EDM-VP and report the performance of adversarial images in Table 12.

Table 12: We report the performance of EvoSeed with Stochastic Sampling in EDM-VP (Karras et al., 2022) Diffusion Model $G$.

| Classifier Model $F$ | $\varepsilon = 0.3$ | $\varepsilon = 0.2$ | $\varepsilon = 0.1$ |
|---|---|---|---|
| Standard Non Robust (Croce et al., 2021) | 92.4% | 85.7% | 75.8% |
| Corruptions Robust (Diffenderfer et al., 2021) | 89.2% | 83.7% | 65.9% |
| $L_2$ Robust (Wang et al., 2023b) | 98.4% | 93.9% | 68.6% |
| $L_\infty$ Robust (Wang et al., 2023b) | 99.4% | 94.9% | 76.8% |

Table 13: We report the performance of EvoSeed with Differential Evolution as Optimizer.

| Classifier Model $F$ | $\varepsilon = 0.3$ | $\varepsilon = 0.2$ | $\varepsilon = 0.1$ |
|---|---|---|---|
| Standard Non Robust (Croce et al., 2021) | 79.7% | 62.6% | 38.7% |
| Corruptions Robust (Diffenderfer et al., 2021) | 70.9% | 49.6% | 22.8% |
| $L_2$ Robust (Wang et al., 2023b) | 69.2% | 49.6% | 23.5% |
| $L_\infty$ Robust (Wang et al., 2023b) | 73.6% | 69.2% | 32.6% |

Table 14: We report the performance of EvoSeed in a white-box setting using PGD Backpropagation optimization.

| Classifier Model $F$ | NFE = 10 | NFE = 100 | NFE = 2800 |
|---|---|---|---|
| Standard Non Robust (Croce et al., 2021) | 54.0% | 98.0% | 100.0% |
| Corruptions Robust (Diffenderfer et al., 2021) | 50.0% | 97.0% | 100.0% |
| $L_2$ Robust (Wang et al., 2023b) | 67.0% | 94.0% | 100.0% |
| $L_\infty$ Robust (Wang et al., 2023b) | 75.0% | 100.0% | 100.0% |

### F.4 ANALYSIS OF PERFORMANCE OF EVOSEED WITH DIFFERENTIAL EVOLUTION

To understand the effect of using other black-box optimizer in our EvoSeed Framework, we experiment with Differential Evolution (Storn & Price, 1997) and report the performance in Table 13. We also find that other black-box optimizers like SPSA (Spall, 1992), L-BFGS-B (Liu & Nocedal, 1989), and TNC (Martens et al., 2010) did not converge, while other optimizers like DIRECT (Gablonsky & Kelley, 2001) have significantly more number of function evaluations than CMA-ES. Since gradient-based black-box optimizers fail to converge, we observe that estimating gradients around the seed vector (initial or pseudo-optimized) is challenging because nearby seed vectors often yield similar function evaluations, leading to insignificant gradient estimation. This similarity in function evaluations hinders the convergence of gradient-based optimization methods. In contrast, evolution-based optimization efficiently explores the search space by significantly altering the seed vector. Simultaneously, it exploits by evolving new generations of the seed vector around the pseudo-optimal seed vector to enhance optimization.

### F.5 ANALYSIS OF PERFORMANCE OF EVOSEED FRAMEWORK IN A WHITE-BOX SETTING

We also experiment generating adversarial samples in a white-box setting using PGD Backpropagation optimization on the initial seed vector $z$ as reported in Table 14. We note that access to the model weights can increase the efficiency of the EvoSeed by approximately $28\%$. As a white-box variant of EvoSeed with NFE = 100 has similar performance to the black-box variant of EvoSeed with NFE = 2800. This also shows that black-box variant fares reasonably well compared with the white-box variant, noting that the black-box does not have access to model parameters, which is a significant advantage for the white-box attacks.

### F.6 ANALYSIS OF IMAGES GENERATED OVER THE GENERATIONS

Here, we analyse the EvoSeed's performance with respect to the number of generations, as shown in Figure 16. We observe that, for EvoSeed with $\varepsilon = 0.1$, the curves do not saturate suggesting that

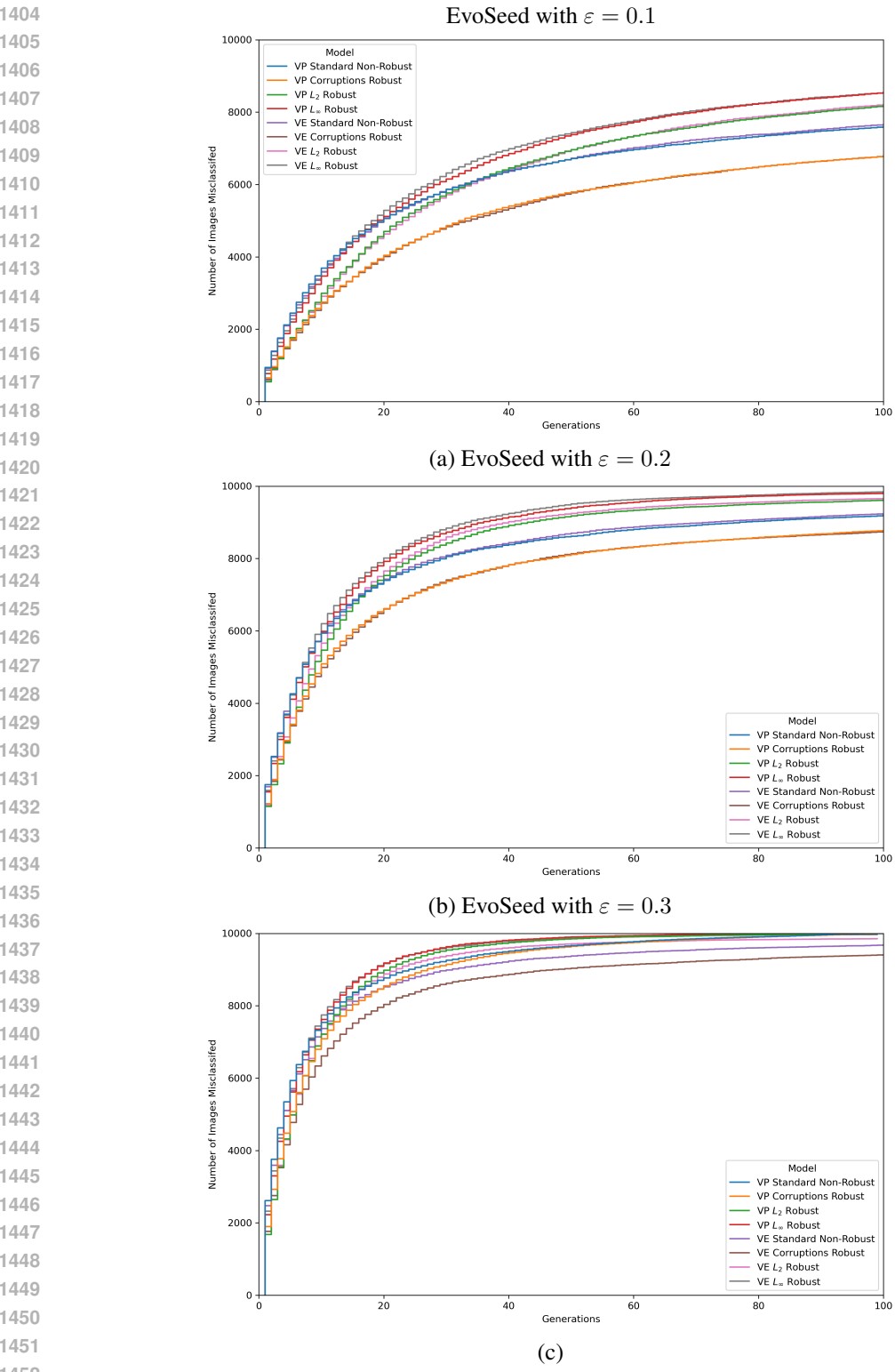

(a) EvoSeed with $\varepsilon = 0.2$

(b) EvoSeed with $\varepsilon = 0.3$

(c)

Figure 16: Accuracy on Generated Images $x$ by the classifier model $F$ over $\tau$ generations. (a) compares the performance of EvoSeed and RandSeed, while (b) compares the performance of EvoSeed with different classifier models.

a higher number of generations to craft natural adversarial samples will further improve the attack performance.

