# OpenReview forum: "Breaking Free: Hacking Diffusion Models for Generating Adversarial Examples and Bypassing Safety Guardrails"
_ICLR.cc/2025/Conference — Submitted to ICLR 2025_

### Official Review · Reviewer_xR8k · 2024-10-27

**Soundness:** 3
**Presentation:** 3
**Contribution:** 1
**Rating:** 3
**Confidence:** 5

**Summary:**

This paper presents EvoSeed, an evolutionary strategy-based framework for generating adversarial samples. EvoSeed integrates a conditional diffusion model, a classifier model, and a seed-searching module using CMA-ES. Experimental results demonstrate that EvoSeed achieves a high attack success rate while maintaining imperceptibility to human observers.

**Strengths:**

1, Modifying the seed, rather than the image generation model (in this case, a conditional diffusion model), is an intriguing approach.

2, The proposed model demonstrates performance across a wide range of tasks.

3, The paper is well-organized and easy to follow.

**Weaknesses:**

1, The comparison with previous methods lacks rigor. The paper places excessive focus on comparing performance across different application areas without explaining the technical differences among them. The rationale for modifying the seed rather than the generative model, along with any demonstrated improvements over previous methods, is largely absent. Furthermore, comparing EvoSeed to only a single method is insufficient to substantiate its effectiveness.

2,  In line 350, the authors state that EvoSeed can serve as a tool for understanding misclassification spaces, citing an example where the confidence in identifying a volcano image drops from 0.81420 to 0.01745 as the smoke and fire areas diminish, resulting in misclassification (Figure 7). However, there's no clear evidence that this drop in confidence is due to the reduced smoke and fire areas; it could just as easily be attributed to invisible texture changes—a common factor in adversarial attacks.

**Questions:**

1, What are the primary benefits of the proposed method (modifying the seed) compared to the more common approach of modifying the generative model?

2, What distinguishes the tasks presented in Sections 4.1, 4.2, and 4.3 within the context of an adversarial attack generation framework? Why do you believe these tasks warrant separate discussions?

---

### Official Review · Reviewer_Jamt · 2024-10-28

**Soundness:** 3
**Presentation:** 3
**Contribution:** 2
**Rating:** 3
**Confidence:** 4

**Summary:**

This paper introduces EvoSeed, an evolutionary strategy-based algorithm for generating natural adversarial examples using conditional diffusion models. The generated adversarial samples appear photorealistic, evading human perception while misleading classifiers across multiple tasks. EvoSeed presents new challenges by bypassing safety mechanisms, such as NSFW filters and commercial APIs.

**Strengths:**

* EvoSeed leverages CMA-ES optimization to refine the initial seed vector, enabling high-quality adversarial images.
* The framework demonstrates misclassification across various tasks, showing versatility in both attacks and system diagnostics.
* EvoSeed offers value as a diagnostic tool to probe classifier weaknesses, aiding in understanding misclassifications and enhancing robustness testing.
* Metrics such as Attack Success Rate (ASR) and Fréchet Inception Distance (FID) provide strong evidence for EvoSeed’s effectiveness in generating adversarial examples.

**Weaknesses:**

Major
* EvoSeed relies on an iterative optimization process, making it significantly slower than other adversarial attacks. Furthermore, the adversarially generated images differ substantially from the original generated images, and in some cases, they appear unnatural. This issue arises because EvoSeed does not constrain changes in the pixel space, unlike other adversarial methods that impose norms such as L_infty  or L_2  to limit perturbations. An example of this unnatural behavior can be seen in Figure 8 with the shovel/panda image.
* The paper lacks a comparison with a simple two-step baseline, where an image is first generated using a standard diffusion model and then attacked using a traditional adversarial attack. This baseline would help establish whether EvoSeed offers meaningful improvements over this simpler and more efficient method. Both methods should operate under the same pixel-space threat model, ensuring that any advantages of EvoSeed are fairly evaluated
* EvoSeed shares strong similarities with Blau et al [1] approach, which uses diffusion models for adversarial defense by optimizing latent variables with perturbation constraints. A comparison in the related work and results sections is required to demonstrate how EvoSeed offers new insights or improvements. Both methods employ latent-space optimization while restricting the allowed perturbation, making this comparison essential for understanding EvoSeed’s contribution.
* The quantitative evaluation relies solely on CIFAR-10 and MNIST, which limits the generalizability of the results. Including more diverse datasets would provide a stronger foundation for the claims made.
* EvoSeed applies a perturbation norm of ε=0.3, which is ten times larger than the standard value of 8/255≈0.03 commonly used for CIFAR-10. This large perturbation raises concerns about the fairness of comparisons, even if it is in the latent space. Furthermore, it is essential to include a distance metric to measure how different the adversarially generated image is from the original generated one, as this would provide a clearer understanding of the magnitude of changes introduced by the attack.


Minor
* On line 87, the authors state that the generated samples come from the image distribution, but this claim is not substantiated. Some generated images appear unnatural, such as the shovel/panda example in Fig 8.
* On line 355, the authors suggest that standard adversarial attacks do not provide explainability. However, prior research (e.g., Etmann et al., 2019 [2]) has demonstrated that adversarially trained classifiers possess perceptually aligned gradients, offering some level of interpretability

[1] Blau, Tsachi, et al. "Threat model-agnostic adversarial defense using diffusion models." arXiv preprint arXiv:2207.08089 (2022).
[2] Etmann, Christian, et al. "On the connection between adversarial robustness and saliency map interpretability." arXiv preprint arXiv:1905.04172 (2019).

**Questions:**

See weaknesses

---

### Official Review · Reviewer_Qy1Q · 2024-10-31

**Soundness:** 2
**Presentation:** 3
**Contribution:** 2
**Rating:** 5
**Confidence:** 3

**Summary:**

This paper proposes EvoSeed which uses a diffusion model to generate natural adversarial examples. These examples can induce misclassifications in classifiers across various task scenarios, including object classification and safety content detection. The paper validates the effectiveness of EvoSeed in generating adversarial examples through qualitative analysis of the generated examples and quantitative experiments.

**Strengths:**

1. The writing is good, with clear descriptions of the content.
2. It validates the attack effectiveness of natural adversarial examples across multiple classification task scenarios.
3. The paper provides a qualitative analysis of the various phenomena exhibited by the generated natural adversarial examples from multiple perspectives.

**Weaknesses:**

1. Novelty: There have been prior works [1-3] using diffusion models to generate natural adversarial examples. The authors need to clearly articulate the novelty of their approach compared to these existing works.

2. Lack of Comparative baselines: The paper lacks a comparison of its results with similar works [1-3] that also utilize diffusion models to generate natural adversarial examples. It is important to include these methods as baselines to evaluate the attack effectiveness of the generated adversarial samples.
[1] Dai, Xuelong, Kaisheng Liang, and Bin Xiao. "Advdiff: Generating unrestricted adversarial examples using diffusion models." European Conference on Computer Vision. Springer, Cham, 2025.
[2] Xu, Kangze, et al. "Transferable and high-quality adversarial example generation leveraging diffusion model." 2024 IEEE International Conference on Multimedia and Expo (ICME). IEEE, 2024.
[3] Chen, Xinquan, et al. "Advdiffuser: Natural adversarial example synthesis with diffusion models." Proceedings of the IEEE/CVF International Conference on Computer Vision. 2023.

3. The rationale behind certain experimental setup choices is unclear:
1）Selection of Diffusion Models: In sections 4.1, 4.3, and 5.1, different diffusion models are employed in the experiments. The authors need to explain the reasons for selecting these specific models.
2）Choice of Victim Models: The authors should clarify the criteria for selecting the victim models used in different task scenarios.

4. Some conclusions require experimental validation:
1）In the quantitative experiments, the authors only validate their approach on the object classification task. It would be beneficial to provide quantitative results for other tasks mentioned in the paper, such as safety detection.
2）The claims made in section 4.1, stating that "our method outperforms adversarial image generation using Text-to-Image Diffusion Models like Liu et al. (2024b) and Poyuan et al. (2023), which disrupt the alignment with the conditioning prompt c," and in section 4.2, which asserts that "Schramowski et al. (2023) provides prompts to bypass these classifiers; however, we use simple prompts that effectively generate inappropriate images," require validation through quantitative experiments.
3）The authors state in the abstract, "Our research opens new avenues for understanding the limitations of current safety mechanisms." However, there is already a substantial body of research on the safety detection mechanisms of text-to-image models. The authors should consider comparing their findings with these existing works[4,5,6].
[4] Qu, Yiting, et al. "Unsafe diffusion: On the generation of unsafe images and hateful memes from text-to-image models." Proceedings of the 2023 ACM SIGSAC Conference on Computer and Communications Security. 2023.
[5] Yang, Yuchen, et al. "Sneakyprompt: Jailbreaking text-to-image generative models." 2024 IEEE symposium on security and privacy (SP). IEEE, 2024.
[6] Ba, Zhongjie, et al. "SurrogatePrompt: Bypassing the Safety Filter of Text-To-Image Models via Substitution." arXiv preprint arXiv:2309.14122 (2023).

**Questions:**

1. How does the attack effectiveness of this method compare to that of previous approaches?
2. How were the models chosen in the experiments determined, including both the diffusion model and the victim models for each task?
3. What advantages does this method have compared to existing attacks on text-to-image model safety checkers?

---

### Meta-Review · Area_Chair_y7Vm · 2024-12-17

**Metareview:**

This paper introduces EvoSeed, a novel framework that uses CMA-ES and conditional diffusion models to generate natural adversarial examples that mislead classifiers while maintaining perceptual quality. The reviewers generally found the concept interesting but raised significant concerns about novelty, comparisons to prior works, and the lack of experimental design; and they thus uniformly lend toward rejection. I think this work would benefit from a significant revision and then be resubitted to a future venue.

**Additional Comments On Reviewer Discussion:**

The authors did not address the concerns described above through a rebuttal and no further discussion was raised among the reviewers as well.

---

### Decision · Program_Chairs · 2025-01-22

Reject